# Acoustic Emission-Based Analysis of Damage Mechanisms in Filament Wound Fiber Reinforced Composite Tubes

**DOI:** 10.3390/s23156994

**Published:** 2023-08-07

**Authors:** Parsa Ghahremani, Mehdi Ahmadi Najafabadi, Sajad Alimirzaei, Mohammad Fotouhi

**Affiliations:** 1Faculty of Mechanical Engineering, Amirkabir University of Technology (Tehran Polytechnic), 424 Hafez Ave, Tehran 15914, Iran; p.ghahremani@aut.ac.ir (P.G.); ahmadin@aut.ac.ir (M.A.N.); alimirzaei69@aut.ac.ir (S.A.); 2Faculty of Civil Engineering and Geosciences, Delft University of Technology, 2628 CN Delft, The Netherlands

**Keywords:** composite tube, strip patch, damage mechanisms, acoustic emission, filament winding

## Abstract

This study investigates the mechanical behavior and damage mechanisms of thin-walled glass/epoxy filament wound tubes under quasi-static lateral loads. The novelty is that the tubes are reinforced in critical areas using strip composite patches to provide a topology-optimized tube, and their damage mechanisms and mechanical performance are compared to that of un-reinforced (reference) tubes. To detect the types of damage mechanisms and their progression, the Acoustic Emission (AE) method is employed, accompanied by data clustering analysis. The loading conditions are simulated using the finite element method, and the results are validated through experimental testing. The findings confirm that the inclusion of reinforcing patches improves the stress distribution, leading to enhanced load carrying capacity, stiffness, and energy absorption. Compared to the reference tubes, the reinforced tubes exhibit a remarkable increase of 23.25% in the load carrying capacity, 33.46% in the tube’s stiffness, and 23.67% in energy absorption. The analysis of the AE results reveals that both the reference and reinforced tubes experience damage mechanisms such as matrix cracking, fiber-matrix debonding, delamination, and fiber fracture. However, after matrix cracking, delamination becomes dominant in the reinforced tubes, while fiber failure prevails in the reference tubes. Moreover, by combining the AE energy and mechanical energy using the Sentry function, it is observed that the reinforced tubes exhibit a lower rate of damage propagation, indicating superior resistance to damage propagation compared to the reference tubes.

## 1. Introduction

Fiber-reinforced polymer (FRP) composites have gained significant attention in various industries—including marine, automotive, construction, and aerospace—due to their unique properties. These materials possess exceptional advantages such as a high strength-to-weight ratio, corrosion resistance, and high durability, which make them an ideal choice for many engineering applications. In addition, the high energy absorption capacity of FRP composites is attributed to their extensive damage mechanisms and controlled progressive degradation process [1]. The utilization of FRP composites also offers cost-saving benefits in terms of reducing the production and installation costs and minimizing the expenses associated with repairs and maintenance [2]. Consequently, the adoption of composite structures has been widespread, which necessitates a proper investigation of their behavior under different loading conditions.

Among the different FRP components, composite tubes have gained attention due to their corrosion resistance, lightweight, and cost-effective performance. These composite tubes are subjected to lateral loads and their performance is affected by their material properties, stacking sequence, geometry, and type of damage mechanisms induced during their service. Therefore, it is crucial to determine the strength and damage mechanisms of these composite tubes from an engineering perspective [3]. In a study by Zhang et al. [4], composite structures of various shapes and sizes were investigated, and it was concluded that a hollow cylinder with a circular cross-section exhibits the highest specific energy absorption compared to other shapes, such as cones or square-profiles. Özbek et al. [5] investigated the effect of different fiber orientations and types (glass and carbon fibers in hybrid form) on the load-bearing capacity and energy absorption capability of polymer matrix composites under quasi-static lateral loading. It was found that using a combination of glass and carbon fibers improves the structure’s energy absorption and load-bearing capacity. Sun et al. [6] studied the impact of the stacking sequence on the transverse energy absorption of tubes made of aluminum and Carbon fiber-reinforced polymer (CFRP). Their findings indicated that the hybrid structure exhibited a superior energy absorption capacity compared to the individual aluminum and CFRP structures. Zhu et al. [7] conducted finite element modelling (FEM) to investigate the behavior of CFRP structures with different cross-sectional geometries under compression. Their results showed that cylindrical tubes have a higher energy absorption capacity compared to the other samples.

Other researchers have conducted studies on the failure process and types of damage mechanisms in composite tubes. Alimirzaei et al. [8,9] investigated the damage mechanisms in filament-wound composite tubes under axial compressive loading. The experimental test results showed that the primary damage mechanism in composite tubes was local fiber breakage accompanied by buckling. Eggers et al. [10] analyzed the dominant damage mechanisms and the effect of some filament winding process parameters on the behavior of carbon/epoxy rings under radial compression, axial compression, and hoop tensile loadings. The dominant damage mechanisms were delamination, delamination and minor off-axis cracks, and fiber/matrix debonding along with fiber breakage, respectively. Almeida et al. [11] proposed a new model for investigating the radial behavior and damage mechanisms of different filament-wound composite tubes under pressure. The numerical results demonstrated that the tubes possessing a diameter-to-thickness ratio (*d/t*) below 20:1 fail due to buckling, whereas the tube with a higher *d/t* ratio predominantly exhibits damage induced by in-plane shear, leading to delaminations. Dadashi and Rahimi investigated the damage mechanisms’ initiation and growth in composite cylinders under lateral compression [12]. Pavan et al. [13] developed a material model that accounted for viscoelastic effects in the composite damage process using continuum damage mechanics. Rafiee et al. [14,15] developed a computational model based on empirical and theoretical approaches to predict the compressive and hoop tensile behavior of glass fiber-reinforced polymer composite tubes. The observation revealed that as the fiber volume fraction increases, in-plane damage is more likely to occur in the outermost layer of the FRP at lower levels of diametric deflection as a result of the reduction in the transverse strength. In contrast, increasing the winding angle will postpone delamination at the interface of the core layer and adjacent cross ply.

Acoustic emission (AE) monitoring is an effective method for identifying the main damage mechanisms in composites. There has been significant research in recent years on the use of AE monitoring for damage mechanisms to detect and predict the behavior of composite materials [16,17]. For instance, Boussetta et al. [18] investigated the AE activities in longitudinally cut strips of a filament-wound tube under tension testing. Then, unsupervised pattern recognition analysis was used to process the AE signals, and the Kohonen self-organizing map was found to be more efficient than other pattern recognition methods. Šofer et al. [19] used the AE method to identify damage mechanisms in CFRP pipes subjected to three-point bending testing. Ghasemi et al. [20] employed the AE method to classify different damage mechanisms in laminated composites under tensile loading using Wavelet Packet Transform and fuzzy clustering. In another study, Fotouhi et al. [21] classified the damage mechanisms in woven and unidirectional glass/epoxy samples under three-point bending loading using the semi-supervised fuzzy C-means algorithm. Fiber breakage was the dominant damage mechanism in unidirectional samples, whereas matrix cracking was the most significant damage mechanism in woven samples.

The repair and reinforcement of composite structures has become a significant concern for engineers as these structures may become structurally weak over time due to design errors, poor construction quality, exposure to harsh environments, or additional loads. One way to locally reinforce a structure is by installing patches in critical areas of a component. Bhatia et al. [22] conducted an experimental investigation on the static and fatigue behavior of repaired laminates using strip patches under bending loads. Three types of configurations—including patch bonding in tension, compression, and double-sided—were examined, and it was observed that the double-sided patch configuration performed better than the one-sided patch configuration. Andrew and Arumugam [23] studied the behavior of glass/epoxy samples repaired with hybrid composite patches of glass and Kevlar at different ratios under tensile testing. The development of the damage process of the repaired samples was also investigated using the AE monitoring technique. Yoo et al. [24] experimentally studied the strength and damage mechanisms of patched composite plates under static and fatigue tensile loading. They carefully considered various parameters such as the overlap length, patch orientation angle, and damage mechanisms size. It was found that the orientation angle of the patch had the greatest influence on the damage mechanism. Kabir et al. [25] experimentally and numerically investigated steel hollow circular section beams reinforced with CFRP sheets under four-point bending. They tested several layering configurations for the reinforcing cover and recorded the mid-span displacement and load-bearing capacity of the structure.

To date, there has been no significant research on the use of reinforcing composite tubes, and any reinforcements were used only after the damage mechanisms and for repair purposes. In this research, for the first time, the local reinforcement of stress-concentrated and critical areas of composite tubes is conducted by employing strip patches during the manufacturing stage to provide a topology-optimized tube that can better distribute the applied stress. The primary objective of this study is to analyze and compare the behavior of reinforced and conventional (reference) filament wound tubes when subjected to quasi-static lateral loads. The effect of reinforcement on the maximum load-carrying capacity and stiffness of the tubes was investigated. Experimentally validated FEM was used to simulate the behavior of the reference tube and to identify the critical regions requiring reinforcement, as well as to evaluate the stress distribution. Furthermore, the study aims to examine the initiation and growth of different types of damage mechanisms that occur during the loading process. AE testing was utilized to detect the mechanisms of damage and their growth in the composite tubes. This method enabled the detection of the onset of damage, and different damage mechanisms in the samples were identified by analyzing the data using the self-organizing map (SOM) clustering technique.

## 2. Experimental Tests

### 2.1. Sample Fabrication

The filament-wound tubes were made of E-glass fibers with TEX 1200. The sample diameter was considered to be 50 mm. The optimal winding angle for a quasi-statically loaded composite cylinder under lateral compression has been established to be in the range of 55 and 75 degrees [26]. Consequently, a winding angle of 65 degrees was selected for this study. Following the filament-winding process, the tubes were left to dry for 24 h at room temperature. Subsequently, each tube was cut into 50 mm-long rings, and a total of eight samples were produced. To guarantee the homogeneity of the samples, all of the samples were weighed, with the maximum weight difference of the tubes being 3.79%. This weight difference suggests that the manufacturing quality was acceptable.

Studies indicate that the most critical points in a laterally pressurized cylinder are the ones on its left and right sides [12,27]. Therefore, this research focuses on reinforcing these critical areas and analyzing their impact on the cylinder’s structural behavior. One effective method for reinforcing these areas is by applying a strip patch. To ensure a strong bond between the patch and the cylinder, the mechanical and physical properties of the patch material should match those of the base material. As a result, the same fiber reinforcement is often used for the external patch as the fabric used in making the main laminate [23]. A 400 g glass fiber woven fabric was used as the patch, similar to the main fiber material. The efficiency and durability of the patch also depend on several parameters, including the thickness of both the main plate and the patch itself [28]. To prevent excessive bending and early separation due to the flexural anchor, the stiffness ratio (thickness to patch length ratio) should be selected carefully, considering the patch thickness. According to reports, the optimal patch thickness is around 60% of the main plate thickness [22]. In this study, as the thickness of the filament-wound tube wall is around 0.6 mm, the patch thickness was selected as 0.36 mm. Additionally, according to the standard, the most appropriate patch length is between 30 and 50 times the thickness. Therefore, the strip width in the circumferential direction of the cylinder was selected as 12 mm. Figure 1 depicts a schematic of the dimensions of the cylinder and the location of the patches.

According to previous studies, due to the lower stress discontinuity between layers with the same orientation, the occurrence of delamination is less likely in off-axis laminates compared to laminates with different ply orientations [29]. As a result, the patches were placed in a manner that the threads of the fabric were aligned with the winding angle, which is 65 degrees. The strips were first soaked in resin and then placed on both sides of the samples, followed by clamping the samples for one day. Finally, the samples were cured at a temperature of 80 degrees Celsius for four hours. Figure 2 illustrates the various stages of sample fabrication and testing.

### 2.2. Performing Experiments

A tensile testing machine with a five-ton capacity was utilized to apply quasi-static loading conditions. To accommodate the lateral loading condition, two flat plates were employed, designed, and manufactured according to the ASTM D2412 standard. The loading was carried out at a constant speed of 4 mm per minute, while the force values and acoustic signals were recorded throughout the test. The test was terminated upon the observation of the first catastrophic damage to the sample. For recording Acoustic Emission (AE) events, the AEWin software was employed, which has a maximum sampling rate of 40 MHz. Two AE sensors, known as PICO, were used. These sensors are broadband, resonant-type, single-crystal piezoelectric transducers manufactured by Physical Acoustics Corporation (PAC). The sensors had a resonance frequency of 518 kHz and an optimum operating range of 20–750 kHz. The selection of these sensors aligns with the fact that the frequency of the AE signals caused by various damage mechanisms in composite materials is primarily below 500 kHz [21,30]. To ensure proper acoustic coupling between the specimen and the sensor, the surface of the sensor was covered with grease. The AE signal was detected by the sensor and amplified using a 2/4/6-AST preamplifier with a gain selector set to 30 dB. The test sampling rate was 1 MHz, and a resolution of 16 bits was used. The threshold level for AE signal detection was determined by considering the amplitude of noise signals and through trial and error. Waveform parameters such as the peak definition time, hit definition time, and hit lock-out time were set to 200 µs, 800 µs, and 1000 µs, respectively. These values were determined based on pencil lead break tests conducted on the specimens. Prior to the test, the data acquisition system was calibrated using the pencil lead break method for all samples. The calibration procedure involved performing multiple lead breakages at different positions between the AE sensors. The sensors were carefully positioned on the internal face, slightly away from the points located on the horizontal diameter of the cylinder. This placement ensured proximity to the critical areas to minimize attenuation and other effects while preventing damage to the sensors during composite failure.

### 2.3. Basis for Detecting and Identifying Damage Mechanisms

The damage process analysis is based on the examination of the AE data collected to identify the various damage mechanisms that occur and linking them to the localized damage mechanisms [18]. In this study, the SOM technique, which is an unsupervised pattern recognition method, was utilized for signal clustering. This method has been proven to be a suitable and promising approach in various applications, such as structural health monitoring, evaluation of pressure vessels, and fatigue testing [19]. SOM does not depend on the initialization and does not require the predefinition of the clusters’ number. This method uses dimension reduction algorithms and graphical analysis of clusters [23]. Pattern recognition analysis identifies the primary features of a space and linearly transforms them to maximize the distance between classes and minimize the range within classes. The size of these classes depends on the time divisions [19]. Many studies have demonstrated that it is possible to recognize damage mechanisms from AE signals obtained during loading on composite materials using one or more temporal features, particularly signal amplitude. However, in most cases, there is an overlap in the distribution of the signal amplitudes corresponding to different damage mechanisms. Therefore, other studies have focused on analyzing multiple parameters simultaneously.

## 3. FEM

FEM of the tube was constructed in the Abaqus commercial software considering the mesoscopic modelling approach. When the thickness is relatively low, it is acceptable to model a filament wound composite as a laminated composite. The composite tube was modelled as a deformable cylindrical shell in a three-dimensional space, with specific dimensions. Analytical rigid square-shaped shells, with a side length of 70 mm, were created as compressive plates. A reference point was placed on this surface. The mechanical properties of the unidirectional glass/epoxy composite, which were averaged from several sources, can be found in Table 1. The filament wound layer was modelled as two layers angled at +65° and −65° relative to the cylinder axis, each with a thickness of 0.3 mm.

The modelling and analysis of the woven composite patches involved assuming each ply to be a two-layer laminate with a 90-degree phase difference in order to reduce the complexities associated with the modelling. Table 2 shows the mechanical properties of the woven fabric, which are the average values taken from sources [2,30,33].

Nonlinear static analysis is carried out to obtain the in-plane stress components. The determination of the stiffness matrix coefficients in the absence of damage mechanisms is a crucial step in the analysis of structural materials. However, when a damage mechanism occurs, it becomes necessary to incorporate its effects into the structural equations that govern the problem [34]. The stiffness matrix coefficients are impacted by the damage parameters, resulting in a reduction from their initial values. For the purpose of detecting all four different modes of damage, the two-dimensional Hashin damage criterion is employed using the Abaqus software. The damage criteria are presented as Equations (1)–(4).

Fiber tensile damage:
(1)$$d_{ft}^{2}={\left(\frac{\sigma_{11}}{X_{T}}\right)}^{2}+{\left(\frac{\sigma_{12}}{S_{L}}\right)}^{2}$$

Fiber compressive damage:
(2)$$d_{fc}^{2}={\left(\frac{\sigma_{11}}{X_{C}}\right)}^{2}$$

Matrix tensile damage:
(3)$$d_{mt}^{2}={\left(\frac{\sigma_{22}}{Y_{T}}\right)}^{2}+{\left(\frac{\sigma_{12}}{S_{L}}\right)}^{2}$$

Matrix compressive damage:
(4)$$d_{mc}^{2}=\left[{\left(\frac{Y_{C}}{2S_{T}}\right)}^{2}-1\right]\frac{\sigma_{22}}{Y_{C}}+{\left(\frac{\sigma_{22}}{2S_{T}}\right)}^{2}+{\left(\frac{\sigma_{12}}{S_{L}}\right)}^{2}$$
The symbols *d_ft_*, *d_fc_*, *d_mt_*, *d_mc_*, respectively, represent the fiber tensile damage parameter, fiber compressive damage parameter, matrix tensile damage parameter, and matrix compressive damage parameter. The tensile strength in the fiber direction is denoted by *X_T_*, while the compressive strength in the fiber direction is represented by *X_C_*. Similarly, *Y_T_* and *Y_C_* stand for the tensile and compressive strength perpendicular to the fibers. The longitudinal and transverse shear strengths are expressed by *S_L_* and *S_T_*, respectively.

The material stiffness degradation following damage initiation and growth can be described using the equations presented below.
(5)$$C_{11}=\left(1-d_{f}\right)C_{11}^{0}$$
(6)$$C_{22}=\left(1-d_{f}\right)\left(1-d_{m}\right)C_{22}^{0}$$
(7)$$C_{33}=\left(1-d_{f}\right)\left(1-d_{m}\right)C_{33}^{0}$$
(8)$$C_{12}=\left(1-d_{f}\right)\left(1-d_{m}\right)C_{12}^{0}$$
(9)$$C_{13}=\left(1-d_{f}\right)\left(1-d_{m}\right)C_{13}^{0}$$
(10)$$C_{23}=\left(1-d_{f}\right)\left(1-d_{m}\right)C_{23}^{0}$$
(11)$$G_{12}=\left(1-d_{f}\right)\left(1-s_{mt}d_{mt}\right)\left(1-s_{mc}d_{mc}\right)G_{12}^{0}$$
(12)$$G_{13}=\left(1-d_{f}\right)\left(1-s_{mt}d_{mt}\right)\left(1-s_{mc}d_{mc}\right)G_{13}^{0}$$
(13)$$G_{23}=\left(1-d_{f}\right)\left(1-s_{mt}d_{mt}\right)\left(1-s_{mc}d_{mc}\right)G_{23}^{0}$$The two parameters, *d_f_* and *d_m_*, represent the general fiber failure and the general matrix damage mechanism, respectively, which are defined according to Equations (14) and (15). The coefficients *s_mt_* and *s_mc_* are assumed to be 0.9 and 0.5, respectively, in order to control the shear stiffness [34].
(14)$$d_{f}=1-\left(1-d_{ft}\right)\left(1-d_{fc}\right)$$
(15)$$d_{m}=1-\left(1-d_{mt}\right)\left(1-d_{mc}\right)$$

Given that the thickness-to-diameter ratio was less than 0.1, the structure was classified as a thin-walled structure, and the out-of-plane stress components were disregarded. Furthermore, the filament-wound composite tube had a single layer; hence, delamination was not taken into account as a damage mechanism [2].

In this study, the cohesive zone modeling (CZM) technique was utilized to detect composite patch debonding. The cohesive surface approach, in which the adhesion between two adjacent bodies is defined as a contact surface with zero thickness, was employed to define the cohesive zone [15]. The available models for analyzing cohesive elements have a linear elastic region, and the damage expansion region can be linear or exponential. Initially, a high stiffness coefficient (penalty stiffness) is defined to maintain the upper and lower surfaces of the non-cohesive elements within an elastic range. As the loading progresses, once the normal or shear stress in the cohesive zone reaches the interlaminar strength associated with any mode, the stiffness gradually decreases. The area under the stress-strain curve (modes I, II, and III) corresponds to the critical fracture energy, which can be expressed using Equation (16).
(16)$$\tau_{i}=\left\{\begin{array}{cc} K\delta_{i} & \delta_{i}^{max}\leq \delta_{i}^{0}\\ \left(1-d_{i}\right)K\delta_{i} & \delta_{i}^{0}<\delta_{i}^{max}<\delta_{i}^{f}\\ 0 & \delta_{i}^{max}\geq \delta_{i}^{f} \end{array}\right.$$In the above equation, *K* represents the penalty stiffness coefficient of the cohesive zone; *d_i_* is the scalar damage parameter associated with a type of damage mechanism that can be calculated as follows:(17)$$d_{i}=\frac{\delta_{i}^{f}\left(\delta_{i}^{max}-\delta_{i}^{0}\right)}{\delta_{i}^{max}\left(\delta_{i}^{f}-\delta_{i}^{0}\right)},i=1,2,3;\;d_{i}\in \left[0,1\right]$$

The Abaqus software offers two methods for defining damage propagation in traction-separation law: displacement and energy-based criteria. When the energy criterion is chosen, the total fracture energy (fracture toughness) must be inputted. Alternatively, when the displacement criterion is used, the effective displacement at the fracture point should be provided. To determine the fracture energy for different mechanisms, only the area under the stress-displacement curve up to the fracture point needs to be calculated through integration, yielding the final displacements, as given in Equations (18)–(20) [35].
(18)$$\delta_{1}^{f}=2G_{IIIC}/T$$
(19)$$\delta_{2}^{f}=2G_{IIC}/S$$
(20)$$\delta_{3}^{f}=2G_{IC}/N$$The normal strength is represented by *N*, and *S* and *T* denote the shear strengths of the contact surface in the first and second directions, respectively.

As explained earlier, crack initiation occurs when the stress in the cohesive element reaches the strength of the contact surface, and element damage and layer separation occur when the area under the stress-strain curve reaches the fracture toughness or the element deformation exceeds the final displacement [30]. Determining damage initiation at the layer interface under a single type of loading is straightforward and is achieved by comparing the stress components and their critical values. However, in composite structures, interlaminar delamination often occurs in the mixed mode, which complicates the analysis [35]. In this study, the growth of interlaminar delamination in the mixed mode is predicted using the power law, which is defined in Equation (21). It has been established that for epoxy resins, assuming *α* = 1 is sufficiently conservative [35].
(21)$${\left\{\frac{G_{I}}{G_{Ic}}\right\}}^{\alpha }+{\left\{\frac{G_{II}}{G_{IIc}}\right\}}^{\alpha }+{\left\{\frac{G_{III}}{G_{IIIc}}\right\}}^{\alpha }=1$$

For this study, the standard solver was utilized, and the contact between the tube and the rigid plates was defined using a surface-to-surface contact constraint. The slave surface was the cylinder, while two flat plates were selected as the master surfaces. A surface friction coefficient of 0.4 was assumed for the contact. The cohesive zone properties used for the contact definition between the tube and patches were obtained from sources [30,35], and are summarized in Table 3. All of the degrees of freedom for the bottom plate were fixed, and it was considered stationary. For the top plate, all of the degrees of freedom, with the exception of the displacement along the *y*-axis, were set to zero. A velocity of 1 mm per second was then assigned to the top plate’s reference point towards the bottom. The composite tube was meshed with four node shell elements (S4R) independently. In order to accurately model the cohesive zone, the patch and patch placement area elements were set to a size of 0.125 mm. Based on the mesh size analysis, the optimal element dimensions of 0.67 mm were selected for the other sections.

## 4. Results and Discussion

### 4.1. Mechanical Behavior Investigation

The force-displacement diagram can be used to study the mechanical response of materials or structures subjected to specific loading conditions. Figure 3 presents the diagram for the reference samples, where the letter “S” denotes these samples. In contrast, “P” represents the samples reinforced with patches. The curves in the diagram exhibit relatively good conformity between them, but there are some differences in their behavior, which are attributed to defects and issues that occurred during the filament winding manufacturing process.

The maximum load-bearing capacity and stiffness are important parameters to consider in studying the mechanical behavior of pipes under lateral loads. The flexural stiffness is an indicator of the deformation that can be tolerated under transverse loading without structural damage [15]. In this study, to calculate the stiffness of the composite tube, the force value at a displacement equal to 5% of its initial diameter is used, as shown in Figure 4. Moreover, to calculate the absorbed energy during loading, the area under the force-displacement curves is used. For a better comparison of the results, the area under all of the curves up to a displacement equal to 70% of the sample’s initial diameter (35 mm) is considered. Table 4 shows the mechanical parameter values for the samples without patches.

The force-displacement curves and mechanical parameters related to the reinforced samples are depicted in Figure 5 and Table 5, respectively. As shown in the curves, the behavior of similar samples is almost identical, although some differences in the force values can be observed in some samples. For instance, samples P2 and P4 show similar behavior to one another, whereas samples P1 and P3 show similar behavior to each other, especially between loads 125–150 N and displacement 15–20 mm. These differences are mainly attributed to defects and issues that occurred during the manufacturing process and hand-layup process for the reinforcement. One of the significant factors is the distance between the glass fiber roving during winding, resulting in weakness in some parts of the composite sample due to the substantial difference in the width of the roving used. Another factor is the inappropriate distribution of resin in some parts of the sample, which reduces the fiber volume fraction and affects the damage mechanism.

After comparing the behavior of different samples, it was concluded that the reinforced samples exhibited more conformity in the elastic region. This was observed through a comparison of the force-displacement curves and their standard deviation values. The reason for this similarity is that the patches cover weaknesses and structural defects in the critical areas of the tubes. It is re-distributing the stress more uniformly and avoids stress concentration. The investigation of the mechanical behavior of both types of samples indicates that reinforcing the critical areas of the structure with a patch increases the maximum load capacity by an average of 23.25%, the tube’s stiffness by 33.46%, and the absorbed energy by 23.67%. The weight of the reinforced samples is only approximately 1.5% higher than that of the reference samples.

### 4.2. FEM Results

In this section, we compare the behavior of the reference and reinforced tubes using FEM. The average force-displacement curves for both the reference and reinforced samples from the experimental tests are shown in Figure 6, represented by S_m_ and P_m_, respectively. The S_FEM_ symbol corresponds to the reference tube, while the P_FEM_ symbol corresponds to the reinforced tube that resulted from the FEM.

The force-displacement curves’ shape, despite the linear behavior of the material, is noteworthy, and is directly related to the structural geometry and type of loading. Similar studies conducted on metal pipes also observed the same trend for the force-displacement curves [36,37]. The increase in slope and the change in curvature during the experiment is due to the deformation of the tube, causing the areas in contact with the load-applying plates to flatten and to increase the contact area of the structure with the plates. As the pressure is almost constant, the force value increases. The experimental and simulated curves have a relatively good agreement, especially in the elastic region. However, as the deformation of the rings increases and damage initiates, the difference between the experimental and numerical data becomes greater. For both types of samples, the maximum load obtained from the numerical solution method is higher due to structural defects and irregularities in the experimental results, causing the structure not to reach its maximum load-bearing capacity. Nonetheless, the difference between the mechanical properties entered into the software and the actual material properties, as well as the use of a laminate model for designing filament wound tubes, should not be overlooked. Table 6 presents the mechanical parameters related to the simulated models and their comparison with the experimental values.

To justify the mechanical behavior of a structure, it is important to study its deformation process and identify the macroscopic damage growth and damage mechanisms. The deformation history of the tube during the experimental tests and the distribution of Mises stress obtained from the numerical simulation of the reference and reinforced samples can be seen in Figure 7 and Figure 8. The figures reveal that after the start of loading, the circular cross-sectional area of the tube turns into an elliptical shape, increasing the length of the torque arm and the stress due to bending in critical areas. It should be noted that the simulated model behaves similarly to the experiments in terms of deformation, with minor differences arising from manufacturing errors and simplified assumptions in the model. Upon examining the stress contours of both samples, it is evident that at the beginning of the experiment, the stress values at the points where the tube contacts the load-applying plates are higher than the other areas. However, with an increase in the moment of bending, the stress values in the critical areas become much higher than those in the areas in contact with the plates.

The examination of the damage parameters is crucial for detecting the occurrence of damage and its damage mechanism, as explained in Section 3. To achieve this, the stress distribution was examined as the initiation and propagation of damage mechanisms at any point depend on the stress values present at that point. Figure 9 illustrates the σ_x_ (S11) and σ_y_ (S22) stress distributions obtained from the FEM for a reference sample (a) and a reinforced sample (b). The figure divides the cross-section of the cylinders into four hypothetical arcs based on the direction of the stresses (red and blue areas). The upper and lower arcs in contact with the applied load plates are under compression at the outer surface and under tension at the inner surface of the cylinder. By changing the length of these arcs to maintain the cross-section circumference, the two side arcs must have a deformation opposite to the upper and lower arcs, which results in the stress distribution in them to be precisely opposite to the previous state. This finding is in line with similar research results [12].

According to Figure 9b, the addition of a strip patch results in an increase in the value of σ_y_ stress in the tension direction in the middle of the reinforced area due to the increase in thickness at the point of the bending moment. However, the value of this stress decreased in the compression direction at the location of the patch installation. The stress distribution in the critical regions experiences a significant decrease, but at the edges of the patch, the magnitude of σx stress increases, leading to the initiation of interfacial separation. This separation causes the transfer of stress at those points to the filament wound layer, resulting in damage development in the main tube wall.

To analyze the spread of the various damage mechanisms using FEM, the count of elements associated with each damage mechanism type (where the damage parameter for the corresponding mechanism equals one) was determined at various loading stages. Figure 10 and Figure 11 depict the distribution of the Hashin damage parameters at the point of catastrophic damage for both types of samples.

Based on the stress distribution and the composite strength in different directions, the occurrence regions of the four types of damage mechanisms shown in the figures are justifiable. The tensile and compressive damage mechanisms correspond to the regions under tension and compression, as mentioned in the stress distribution discussion. As is evident, the main damage mechanism for both types of samples is the matrix tensile and compressive damage mechanism, which involves wide regions of the cylinder wall. The fiber compressive damage mechanism is the dominant secondary mechanism observed in the internal surface of the critical zones, followed by the fiber tensile damage mechanism in the external surface of these regions. However, the fiber tensile damage parameter has not reached its maximum value of one in both types of samples, indicating that complete destruction has not occurred in this mechanism. The installation of a patch in critical areas leads to a decrease in the value of all the damage parameters at those points; however, their distribution increases at the edges of the patch. These results agree well with the observations obtained from the experimental tests, and demonstrate the effectiveness of the FEM.

In this study, two samples were observed using video recording in order to document the development and progression of the damage, as well as the detachment of the patch. The videos revealed that the first signs of detachment occurred 5 min and 14 s after the start of the experiment, corresponding to a displacement of 20.93 mm. However, according to the FEM results, the detachment of the patch occurred at a displacement of 22.75 mm. In all cases, the initial detachment was observed at the corner of the patch, as it had less contact with the adhesive surface, and therefore a weaker connection than other points along the edge of the patch. Moreover, the stress concentration was higher in the corners. As the crack propagated, the detachment spread along the edge of the patch until it reached the centerline of the cylinder. The experimental and numerical comparison of the patch separation moment is shown in Figure 12. The red circle indicates the initiation point of patch separation in a reinforced sample.

### 4.3. Analysis of AE

The recorded AE events generated from the damage mechanisms during the experiments were collected by two piezoelectric sensors, and the results demonstrated a good agreement between them. On average, the total number of AE signals recorded for the reference samples was twice that of the reinforced samples, indicating a greater degree of active damage mechanisms. This observation is consistent with the lower load-bearing capacity of the reference samples in comparison to the reinforced samples. However, the number of AE signals alone is not a suitable indicator for examining the damage mechanisms in the samples as it does not provide information about the intensity, energy, and source of these signals. Therefore, to accurately detect the damage created at each stage of the loading, it is necessary to separate and classify the AE signals from different damage mechanisms. The amplitude, average frequency, energy, duration, and count of the signals were used to cluster the AE signals using the SOM algorithm [18], and the best outcome, with the least overlap, was achieved by average frequency and amplitude distributions. The clustering results for the reference and reinforced samples can be seen in Figure 13.

The clustering results in Figure 13a show that the data from the four samples were categorized into three clusters. Most of the previous studies concluded that the lowest frequency range is related to matrix cracking and the highest frequency range is related to fiber fracture, and the frequency range of interlaminar delamination is between these two ranges [19,20,21]. As those works were related to a similar type of glass/epoxy laminate, the same concept can be used to analyze the AE results. Taking this into consideration, the first cluster, denoted in red, corresponds to a frequency range of 30 and 150 kHz and is primarily associated with matrix cracking, which is also the dominant frequent range. The second cluster (blue) is identified by a frequency range of 150 and 300 kHz and is related to fiber-matrix debonding and delamination. The third cluster (green) corresponds to a frequency range of 300 and 570 kHz, which is related to fiber breakage. The categorization of damage processes solely based on the AE frequency and clustering results is subject to certain limitations, as there is no definitive and strict boundary separating these processes. However, these results provide a general indication of the frequency range associated with each damage mechanism, offering valuable insights into the overall failure process. While the accuracy of the categorization may not reach 100%, these findings contribute to enhancing our comprehension of the underlying mechanisms involved in structural failure. Figure 13b displays the clustering map for the samples with patches. The number of clusters and their frequency range are similar to those of the reference samples, except that the number of signals in the second cluster, corresponding to delamination, is relatively higher than in the reference samples. The average frequency percentage of the data for each cluster is presented in Table 7.

In both types of samples, cluster 1, which is related to matrix damage, has the highest frequency percentage, confirming the numerical results deduced from Figure 10 and Figure 11. After the test, an observation of the reinforced samples revealed almost no fiber failures within the reinforcing patches themselves, and only the edges of the patch were separated from the tube surface, leading to an increase in the percentage of signals corresponding to cluster 2. Moreover, due to the impregnation of the patch and the placement of an additional resin layer in critical areas, the number of signals corresponding to cluster 1 (matrix cracking) is higher in the reinforced samples. Consequently, the percentage of the frequency of cluster 3 (fiber breakage) decreases with the installation of the reinforcing patch.

Based on the analysis of the damage parameters and the numerical results, it has been determined that matrix damage precedes fiber failure. Therefore, the first acoustic events recorded are related to matrix cracking and the subsequent crack growth, which is indicative of the penetration damage mechanism. These signals have low energy and short durations. On the other hand, fiber failure is associated with high energy events that are mostly observed towards the end of the experiment when the stress values in the fiber direction reach the strength of the composite. The stress contours shown in Figure 9 support this observation. Acoustic signals with average energy levels are also linked to interfacial damage occurring between the fibers and matrix.

Finally, the sentry function was utilized to monitor the advancement of the damage through analyzing the AE data. This function integrates the mechanical and acoustic properties of composites, providing insight into the degree of the damage progression and the residual strength of the structure under different loads. It is calculated as the logarithm of the ratio of strain energy to acoustic energy [31]. Figure 14 displays the sentry function plotted against the force-displacement curve for both a reference sample and a reinforced sample.

The sentry function during the test is composed of four states, as shown in the figure above: (1) An increasing function that indicates the stored strain energy and does not show any specific damage; (2) a sudden drop function, indicating significant damage; (3) a constant function that suggests that the material is resistant to damage expansion; (4) a decreasing function, indicating the loss of load-bearing capacity due to damage development and growth [32,38]. Each of these states is identified on the sentry function graph in the figure. It should be noted that the sentry function becomes evident with the emergence of the initial AE signals, which are associated with microcracks in the matrix. These microcracks cause a decrease in the tube stiffness and are manifested by changes in the slope of the force-displacement curve. The descending portions of the sentry function indicate the initiation and progression of damage, with steeper slopes representing faster rates of damage expansion. Repeated drops and rises correspond to different damage mechanisms [31]. The trend of the sentry function shows good agreement with the force-displacement curve for all of the samples, with regions of changing slope and force drops corresponding to areas of sentry function drops. Some minor declines in the sentry function are noticed prior to a significant decline in the force curve, which is followed by a substantial drop in the sentry function at the time of the force curve drop. This pattern indicates the aggregation and fusion of matrix microcracks, ultimately resulting in significant matrix damage. Following this sharp decline, the sentry function exhibits several gradual decreases, each followed by a significant drop, and this sequence of events persists until the final damage of the sample. The descending trend of the sentry function for reinforced samples has a less steep slope than the reference samples, indicating a slower rate of damage expansion in these samples. The relatively wide constant region signifies that the reinforced structure has good resistance against damage expansion.

According to the results of this research, it can be said that due to the improvement of the load-bearing capacity and the increase in the tube stiffness, the local reinforcement of a thin-walled composite tube that is subjected to lateral load is a suitable alternative to the use of thick-walled tubes; in addition to reducing the weight of the structure, the final cost of production will also be reduced. These findings provide valuable insights for the practical implementation of reinforced thin-walled glass/epoxy filament wound tubes in various industries, such as aerospace, automotive, and construction. The enhanced mechanical performance and resistance to damage offer potential benefits in terms of the structural integrity, durability, and safety. Engineers and designers can utilize this knowledge to optimize the design and manufacturing processes of similar composite structures, leading to more reliable and efficient applications in the field [39,40,41].

## 5. Conclusions

The degradation trend and damage mechanisms of filament wound glass/epoxy tubes were investigated under quasi-static lateral loading using the AE method. Experimentally validated FEM was used to evaluate the critical stress-concentrated regions that require reinforcement; then, the composite strip patches were designed according to the level of stress concentration, and were used to reinforce the critical areas and to investigate their effect on the mechanical behavior, stiffness, and resistance to the damage growth of the tube. To analyze the damage mechanisms, the received AE signals were classified based on the SOM clustering method. The compression process of the reference and reinforced samples was simulated in the Abaqus software and compared with the experimental results. In summary, the results obtained are as follows:Utilizing strip patches to reinforce critical areas of composite tubes is a viable option to augment the load bearing capacity and stiffness without a considerable increase in weight. Furthermore, it enhances the energy absorption capacity and extends the progressive damage mechanisms.The samples reinforced with patches displayed more conformity than the reference samples in the elastic region. This suggests that the attachment of a reinforcing patch can mask potential structural defects or weaknesses caused during manufacturing.The experimental curves and those obtained from the simulation showed relatively good agreement, particularly in the elastic region. However, as the tubes underwent more deformation and damage onset occurred, due to the complex nature of the damage, the difference between the experimental and numerical data increased.The results showed that the installation of a patch in critical areas reduced the amount of Von Mises stress and all of the damage parameters; however, their distribution increased at the edges of the patch.The damage mechanisms were categorized into three clusters based on their spatial distribution, including matrix cracking, fiber-matrix debonding and delamination, and fiber fracture. Both the AE and FEM predicted matrix cracking as the dominant damage mechanism. Delamination and fiber fracture were dominant secondary damage mechanisms in the reinforced and reference samples, respectively.A strong correlation was observed between the trend of the sentry function and the force-displacement curve for all the samples. The decreasing trend of the sentry function for the reinforced samples had a lower slope, indicating that the rate of damage propagation in these samples was lower compared to the reference samples.

## Figures and Tables

**Figure 1 sensors-23-06994-f001:**
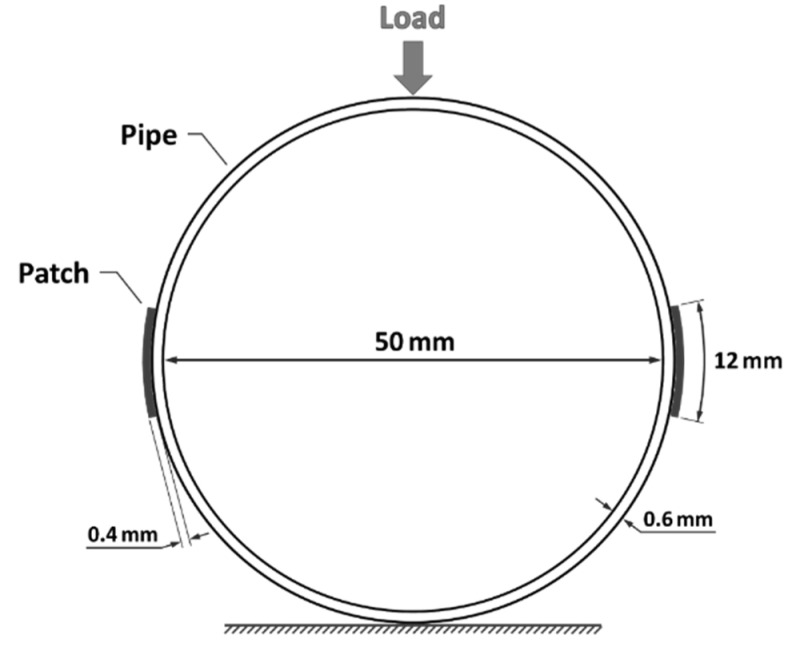
A schematic of the test setup and the location of the reinforcing patches.

**Figure 2 sensors-23-06994-f002:**
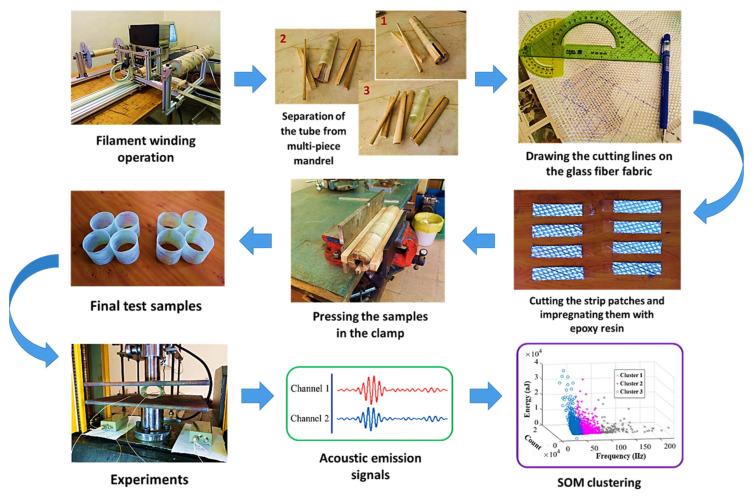
Different steps of the sample preparation and characterization.

**Figure 3 sensors-23-06994-f003:**
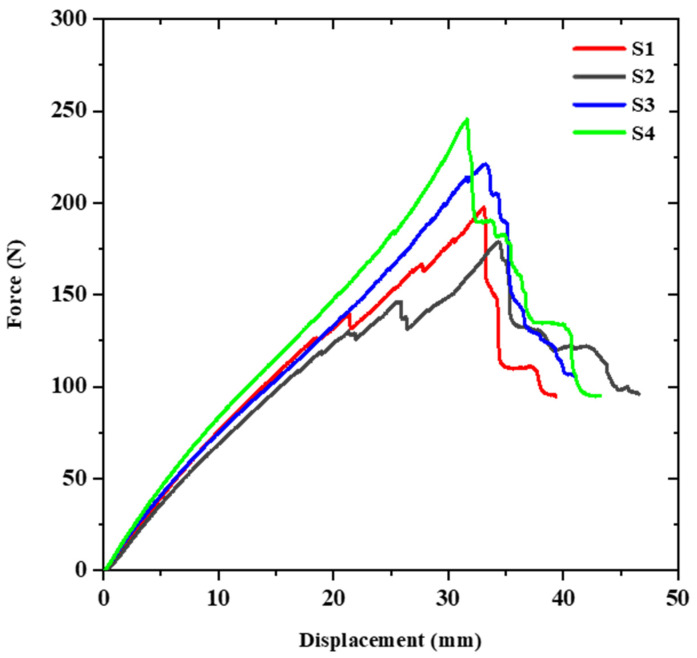
Force-displacement curves of the reference samples.

**Figure 4 sensors-23-06994-f004:**
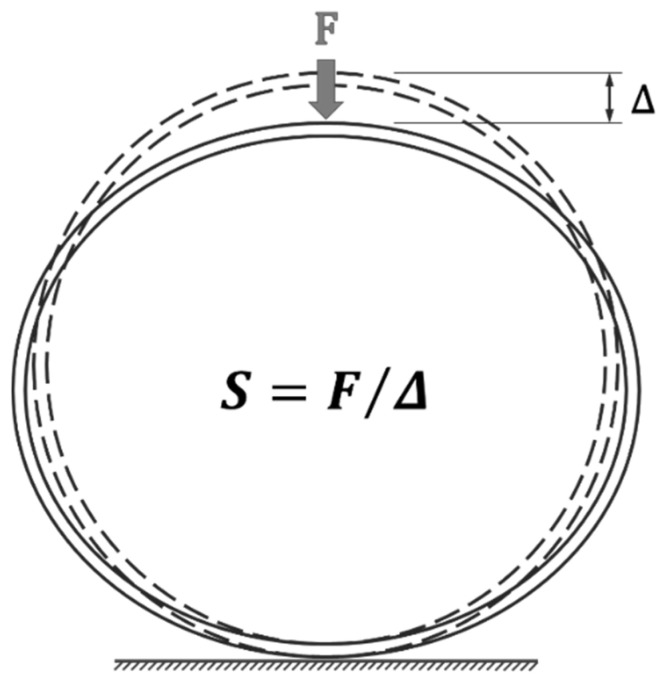
Schematic of how to calculate the bending stiffness of the composite tube.

**Figure 5 sensors-23-06994-f005:**
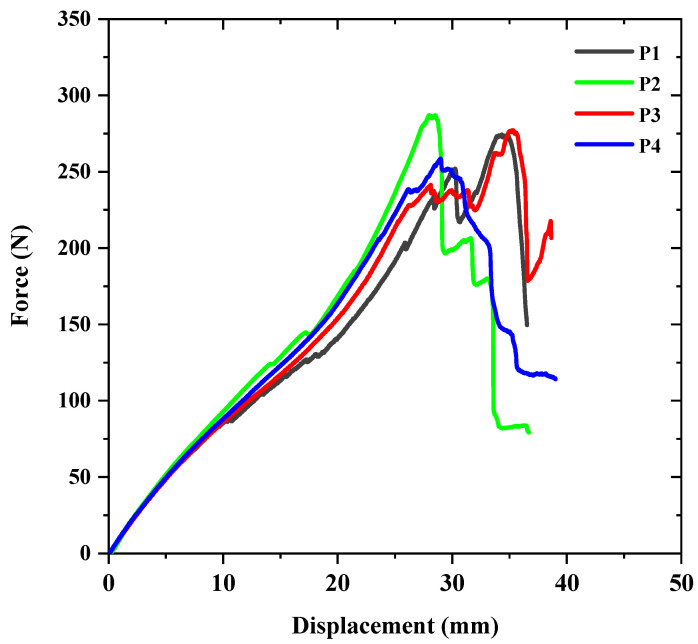
Force-displacement curves of the reinforced samples.

**Figure 6 sensors-23-06994-f006:**
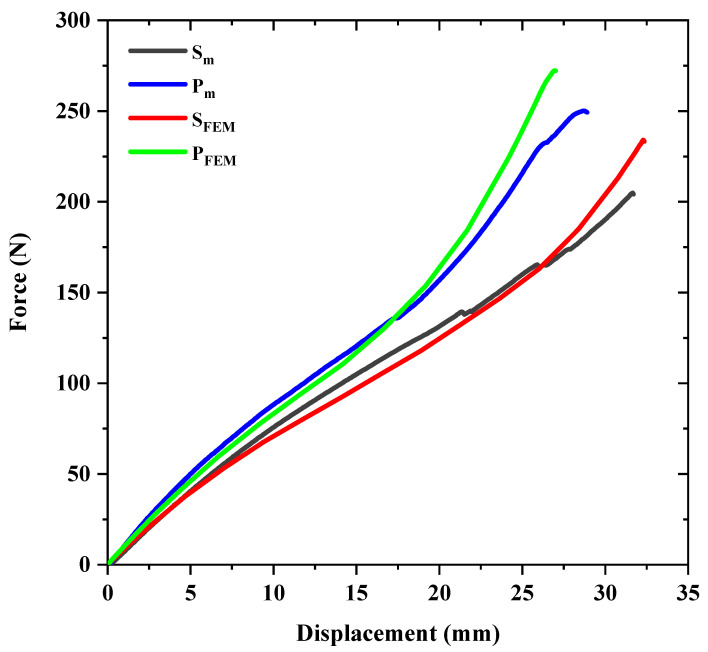
Comparison of the experimental and numerical force-displacement curves for the reference and reinforced samples.

**Figure 7 sensors-23-06994-f007:**
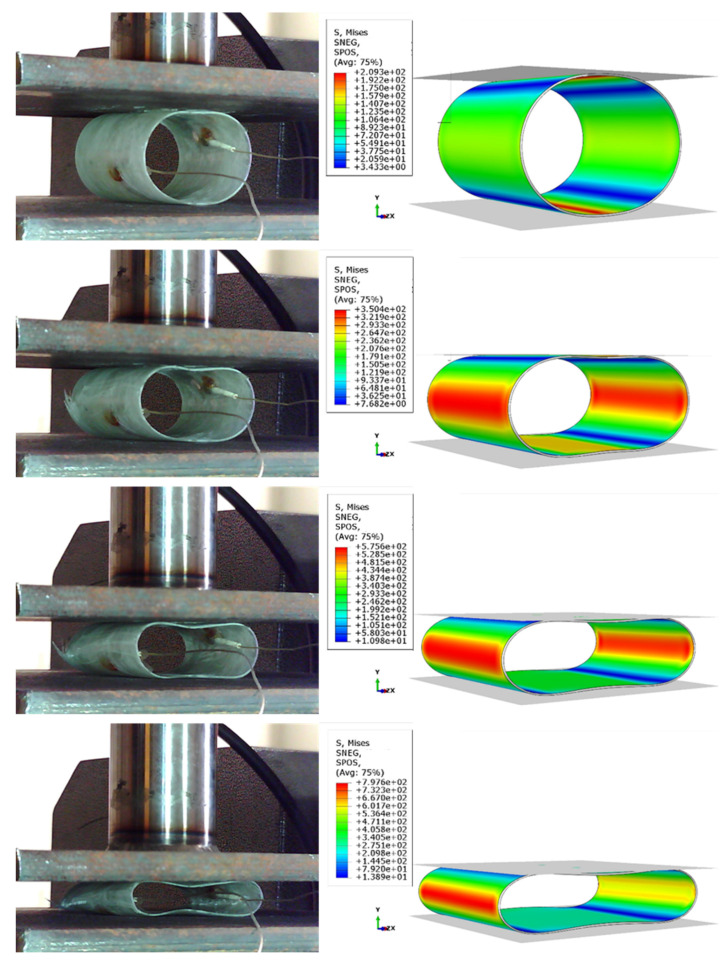
Comparing the deformation process of a reference sample with the FE simulation results.

**Figure 8 sensors-23-06994-f008:**
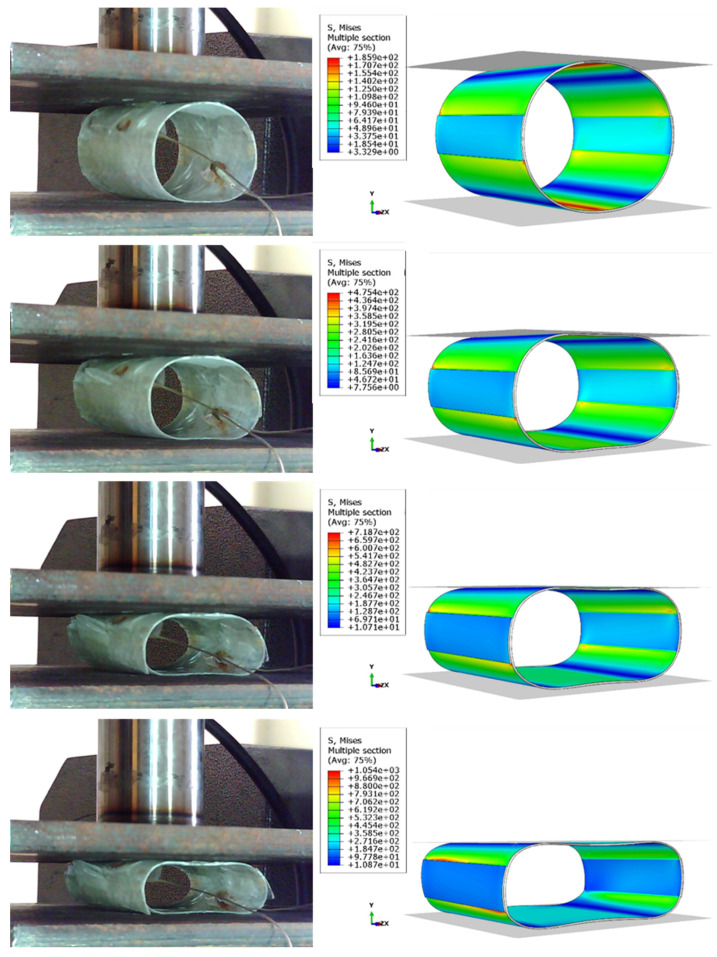
Comparing the deformation process of a reinforced sample with the FE simulation results.

**Figure 9 sensors-23-06994-f009:**
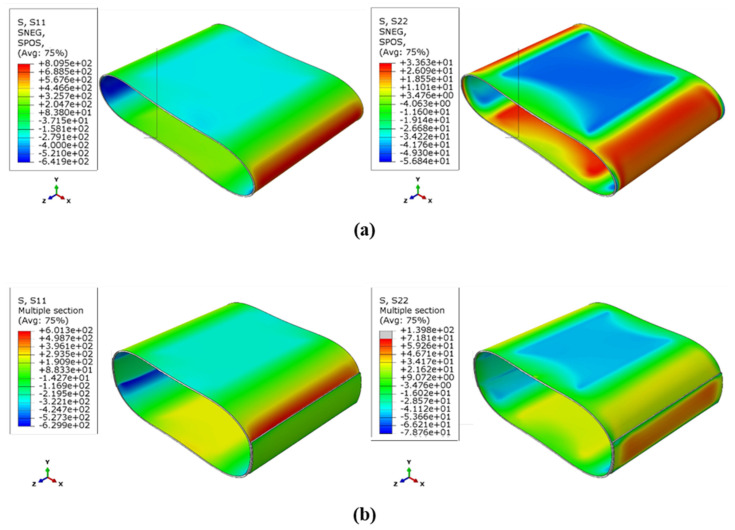
The σ_x_ and σ_y_ stress distributions: (**a**) the reference sample, (**b**) the reinforced sample.

**Figure 10 sensors-23-06994-f010:**
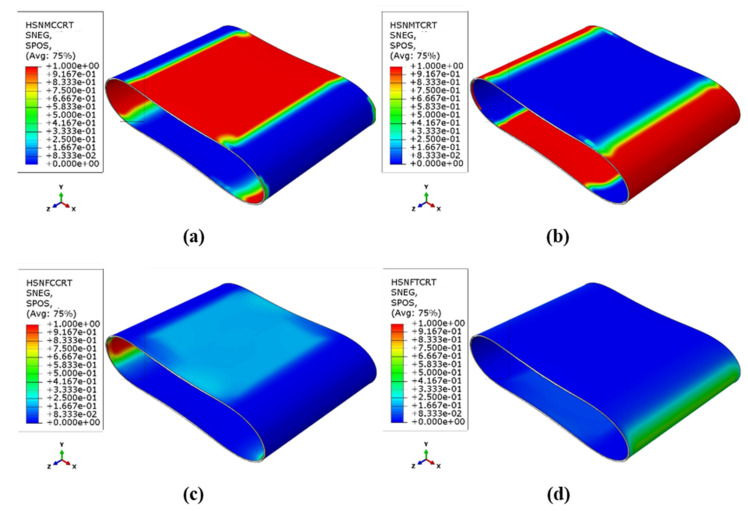
Different Hashin damage parameters in the reference sample: (**a**) Matrix compressive mechanism, (**b**) Matrix tensile mechanism, (**c**) fiber compressive mechanism, (**d**) fiber tensile mechanism.

**Figure 11 sensors-23-06994-f011:**
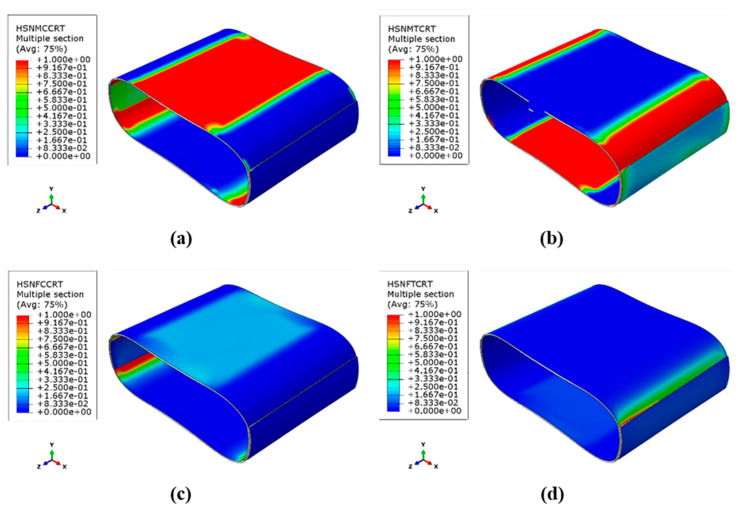
Different Hashin damage parameters in the reinforced sample (**a**) Matrix compressive mechanism, (**b**) Matrix tensile mechanism, (**c**) fiber compressive mechanism, (**d**) fiber tensile mechanism.

**Figure 12 sensors-23-06994-f012:**
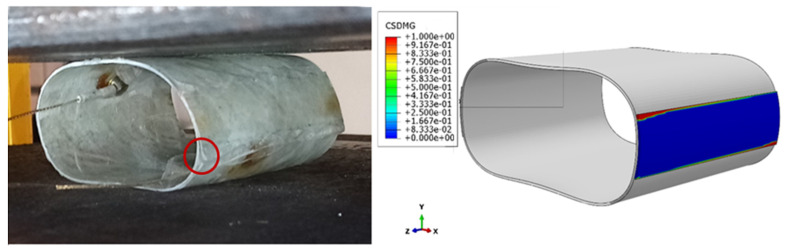
Experimental and numerical comparison of the patch separation moment.

**Figure 13 sensors-23-06994-f013:**
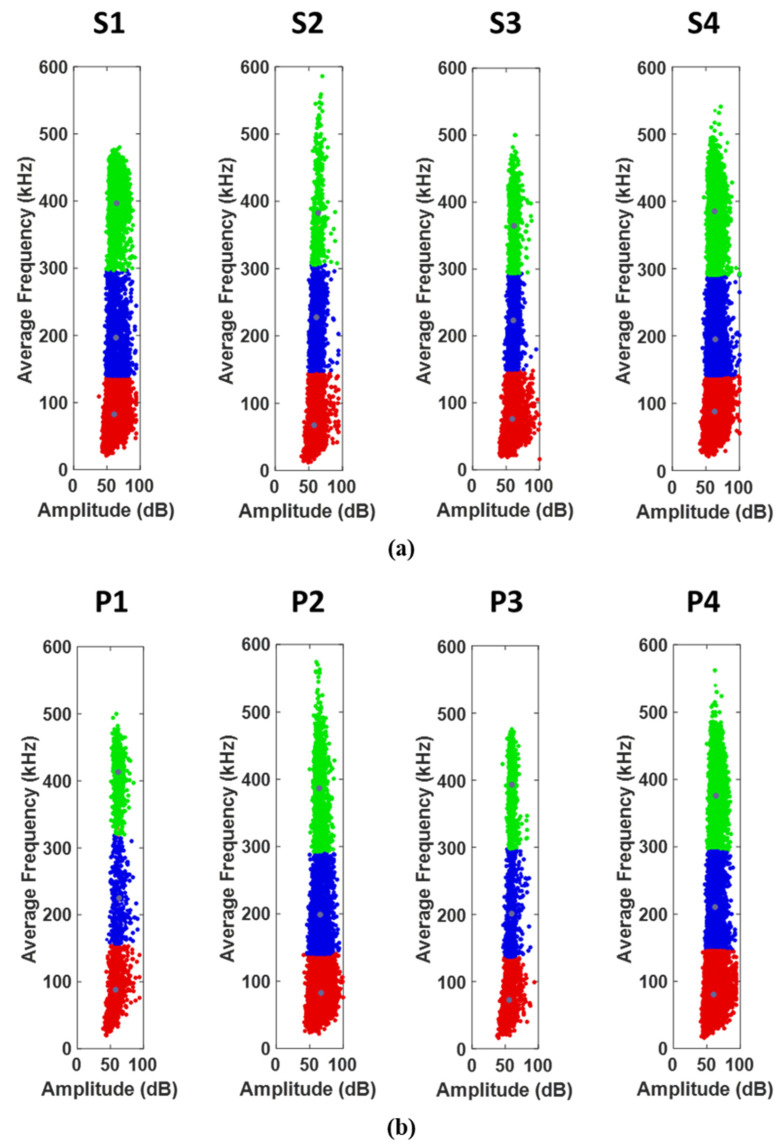
Clustering map: (**a**) for the reference samples, (**b**) for the reinforced samples.

**Figure 14 sensors-23-06994-f014:**
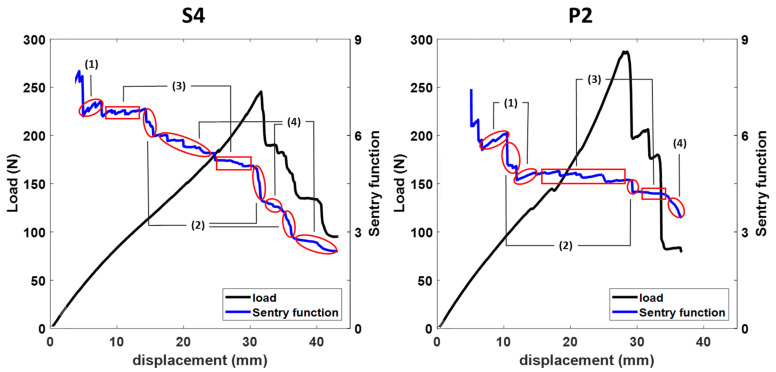
Sentry function graph for a typical reference sample (S4) and reinforced sample (P2).

**Table 1 sensors-23-06994-t001:** Mechanical properties of unidirectional glass/epoxy composite [2,9,30,31,32].

Properties		Values
Elastic Modulus (GPa)	$\mathrm{Longitudinal}\;\mathrm{Young}’s\;\mathrm{modulus}\;E_{11}$	38.7
	$\mathrm{Transverse}\;\mathrm{Young}’s\;\mathrm{modulus}\;E_{22},\;E_{33}$	8.6
	$\mathrm{In}-\mathrm{plane}\;\mathrm{Shear}\;\mathrm{modulus}\;G_{12}$	4.10
	$\mathrm{Out}-\mathrm{of}-\mathrm{plane}\;\mathrm{Shear}\;\mathrm{modulus}\;G_{13},\;G_{23}$	3.25
Poisson’s Ratio	$\mathrm{Longitudinal}\;\mathrm{Poisson}’s\;\mathrm{ratio}\;\nu_{12}$	0.30
	Transverse Poisson’s ratio $\nu_{13},\;\nu_{23}$	0.28
Strength (MPa)	$\mathrm{Longitudinal}\;\mathrm{tensile}\;\mathrm{strength}\;X_{T}$	1064
	$\mathrm{Longitudinal}\;\mathrm{compressive}\;\mathrm{strength}\;X_{C}$	640
	$\mathrm{Transverse}\;\mathrm{tensile}\;\mathrm{strength}\;Y_{T}$	39
	$\mathrm{Transverse}\;\mathrm{compressive}\;\mathrm{strength}\;Y_{C}$	142
	$\mathrm{In}-\mathrm{plane}\;\mathrm{shear}\;\mathrm{strength}\;S_{L}$	51
	$\mathrm{Out}-\mathrm{of}-\mathrm{plane}\;\mathrm{shear}\;\mathrm{strength}\;S_{T}$	89
Fracture Energy (N/mm)	$\mathrm{Longitudinal}\;\mathrm{tensile}\;G_{XT}$	38.4
	$\mathrm{Longitudinal}\;\mathrm{compressive}\;G_{XC}$	19.7
	$\mathrm{Transverse}\;\mathrm{tensile}\;G_{YT}$	3.2
	$\mathrm{Transverse}\;\mathrm{compressive}\;G_{YC}$	4.6

**Table 2 sensors-23-06994-t002:** Mechanical properties of woven fabric glass/epoxy composite.

Properties		Values
Elastic Modulus (GPa)	$\mathrm{Longitudinal}\;\mathrm{Young}’s\;\mathrm{modulus}\;E_{11},\;E_{22}$	22.3
	$\mathrm{Transverse}\;\mathrm{Young}’s\;\mathrm{modulus}\;E_{33}$	8.6
	$\mathrm{In}-\mathrm{plane}\;\mathrm{Shear}\;\mathrm{modulus}\;G_{12}$	5.90
	$\mathrm{Out}-\mathrm{of}-\mathrm{plane}\;\mathrm{Shear}\;\mathrm{modulus}\;G_{13},\;G_{23}$	4.15
Poisson’s Ratio	$\mathrm{Longitudinal}\;\mathrm{Poisson}’s\;\mathrm{ratio}\;\nu_{12}$	0.27
	Transverse Poisson’s ratio $\nu_{13},\;\nu_{23}$	0.36
Strength (MPa)	$\mathrm{Longitudinal}\;\mathrm{tensile}\;\mathrm{strength}\;X_{T}$	492.5
	$\mathrm{Longitudinal}\;\mathrm{compressive}\;\mathrm{strength}\;X_{C}$	284
	$\mathrm{Transverse}\;\mathrm{tensile}\;\mathrm{strength}\;Y_{T}$	492.5
	$\mathrm{Transverse}\;\mathrm{compressive}\;\mathrm{strength}\;Y_{C}$	284
	$\mathrm{In}-\mathrm{plane}\;\mathrm{shear}\;\mathrm{strength}\;S_{L}$	91.6
	$\mathrm{Out}-\mathrm{of}-\mathrm{plane}\;\mathrm{shear}\;\mathrm{strength}\;S_{T}$	91.6

**Table 3 sensors-23-06994-t003:** Mechanical properties of cohesive interface.

Properties	Values
Normal strength of the interface N (MPa)	23
Shear strength of the interface S, T (MPa)	48
Penalty stiffness coefficient$\;K_{nn},\;K_{ss},\;K_{tt}\;$(N/mm^3^)	100,000
Normal fracture toughness $G_{IC}\;$(N/mm)	0.29
Shear fracture toughness $G_{IIC},\;G_{IIIC}\;$(N/mm)	0.70

**Table 4 sensors-23-06994-t004:** Mechanical parameters for the reference samples.

Samples	Displacement Corresponding to Peak Load (mm)	Maximum Load (N)	Stiffness of the Tube (N/mm)	Absorbed Energy (J)
S1	33.08	197.57	7.75	3.61
S2	34.33	178.98	6.39	3.50
S3	33.22	221.18	8.47	4.14
S4	31.64	245.53	9.05	4.47
Average	33.07	210.82	7.92	3.93
SD	0.96	25.01	0.99	0.38

**Table 5 sensors-23-06994-t005:** Mechanical parameters for the reinforced samples.

Samples	Displacement Corresponding to Peak Load (mm)	Maximum Load (N)	Stiffness of the Tube (N/mm)	Absorbed Energy (J)
P1	30.26	251.98	10.54	4.76
P2	28.52	287.02	10.80	4.85
P3	28.13	242.38	10.31	4.93
P4	28.97	258.62	10.66	4.89
Average	28.99	259.85	10.57	4.86
SD	0.80	16.63	0.18	0.06

**Table 6 sensors-23-06994-t006:** Comparison of the FE and experimental results.

Mechanical Parameter	Reference Sample			Reinforced Sample		
	Experimental	FEM	Error Percentage	Experimental	FEM	Error Percentage
Displacement corresponding to peak force (mm)	31.65	32.29	1.98	28.60	26.96	5.73
Maximum load (N)	204.86	233.01	12.07	250.09	272.31	8.16
Stiffness of the tube (N/mm)	7.99	8.38	4.65	9.87	10.68	7.57

**Table 7 sensors-23-06994-t007:** Average frequency of damage mechanisms.

Cluster No.	Reference Samples	Reinforced Samples
1	59.05	57.10
2	16.07	25.85
3	24.88	17.05

## Data Availability

The datasets generated and/or analyzed during the current study are available from the corresponding author on reasonable request.

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
