# Peer review of "Acoustic Emission-Based Analysis of Damage Mechanisms in Filament Wound Fiber Reinforced Composite Tubes"

_sensors, 2023, doi:10.3390/s23156994_

Round 1
Reviewer 1 Report
This paper reported a method to detect the damage mechanisns in filament wound fiber reinforced composite tubes based on the Acoustic emission anaysis. And the finite element modelling and experimental result showed in this paper meet the purpose of research. I would suggest this paper published in this journal after some revises:
1. in Figure 14 and 15, set the y axie in the same range as 0-600 Hz
2. Is it possible to show a figure with Frequency vs Displacement? It would be helpful for reader understanding the acoustic emission analysis when doing the test.
Author Response
- In Figure 14 and 15, set the y axie in the same range as 0-600 Hz
It is modified.
- Is it possible to show a figure with Frequency vs Displacement? It would be helpful for reader understanding the acoustic emission analysis when doing the test.
It would have been good to add the suggested information, however, unfortunately, the raw data source we had is no longer accessible due to some problems in the IT system.
Reviewer 2 Report
I care about the part of AE testing and analysis. Here are some comments for authors to improve the work before final publication.
· Line 183, please provide detailed information on the AE sensor used.
· Line 186, how did the authors determine the threshold as 30 dB? This is a critical point that largely defines the AE signals detected as the “Real” signals from the damage process, rather than noises. During the real test, how to avoid or filter out the signals from the mechanical contact, impact, or friction?
· What’s the setting of PDT, and HDT to define the AE waveforms?
· Line 194, it is recommended that the pencil breaking test is done again after testing, to confirm the reproducibility of the AE sensor response. Figures 8 and 13 show that the specimen deformed largely. Was the coupling of AE sensors ok for testing in the later period of experiments?
· Line 215-216, did the authors analyze other parameters of AE signals? How to choose amplitude and frequency as the key criteria for clustering analysis? Which frequency of AE signals was used as the frequency feature?
· In Figures 14 and 15: were AE features normalized before applying the SOM algorithm for the clustering analysis?
· Getting a reasonable result from the clustering analysis is not easy. But the cluster boundary of the three clusters seems very sharp, even without many overlapping signals. Maybe it is a good clustering result from data science. But if we want to attribute each cluster to a physical process like a specific damage mechanism, it is a bit difficult to say there is a very strict boundary between two damage processes in the response of AE frequency. So, please carefully reconsider the main output of AE results, especially the clustering results.
· Better check the time distribution of each cluster with analysis of the deformation and damage evolution during the whole test.
· It is surprising that no microstructure characterization of fracture or microdamages, which were used to explain the AE signals.
Author Response
- Line 183, please provide detailed information on the AE sensor used.
- Line 186, how did the authors determine the threshold as 30 dB? This is a critical point that largely defines the AE signals detected as the “Real” signals from the damage process, rather than noises. During the real test, how to avoid or filter out the signals from the mechanical contact, impact, or friction?
- What’s the setting of PDT, and HDT to define the AE waveforms?
All the required information is included in the revised manuscript, page 5, lines 196-197. Hit Definition Time (HDT), Peak Definition Time (PDT), and Hit Lockout Time (HLT) were 800 µs, 200 µs, and 1000 µs, respectively.
- Line 194, it is recommended that the pencil breaking test is done again after testing, to confirm the reproducibility of the AE sensor response. Figures 8 and 13 show that the specimen deformed largely. Was the coupling of AE sensors ok for testing in the later period of experiments?
The pencil break test was done before the experiment to assure the proper bonding and calibration. No separation of the sensors was observed during the tests, and the AE results also showed no jump or inconsistency throughout the test that might be associated with the separation of the sensors. The sensors were carefully positioned on the internal face and slightly away from the points located on the horizontal diameter of the cylinder to be close enough to the critical areas to reduce the attenuation and other effects and to prevent damage to the sensors during composite failure.
- Line 215-216, did the authors analyze other parameters of AE signals? How to choose amplitude and frequency as the key criteria for clustering analysis? Which frequency of AE signals was used as the frequency feature?
Yes, other parameters such as energy, duration and count are also considered for clustering, but the best outcome, with the least overlap, was achieved by Frequency and Amplitude distributions. The average frequency used to cluster the data using the SOM algorithm. These are included in the revised version, page 20, lines 476-478.
- In Figures 14 and 15: were AE features normalized before applying the SOM algorithm for the clustering analysis?
No, the real values of the data were used in the SOM algorithm.
- Getting a reasonable result from the clustering analysis is not easy. But the cluster boundary of the three clusters seems very sharp, even without many overlapping signals. Maybe it is a good clustering result from data science. But if we want to attribute each cluster to a physical process like a specific damage mechanism, it is a bit difficult to say there is a very strict boundary between two damage processes in the response of AE frequency. So, please carefully reconsider the main output of AE results, especially the clustering results.
We agree with the reviewer that the connection between the damage mechanisms and clustering results is not 100% accurate, this is discussed in the revised paper. Lines 492-498
- Better check the time distribution of each cluster with analysis of the deformation and damage evolution during the whole test.
It would have been good to add the suggested information, however, unfortunately, the raw data source we had is no longer accessible due to some problems in the IT system.
- It is surprising that no microstructure characterization of fracture or microdamages, which were used to explain the AE signals.
The correlation was done considering previous studies in laminated composites where high amplitude and Frequency signals represent fibre fractures while medium values mostly correspond to delamination and interfacial debonding [3, 18-21]. Since those works were related to a similar type of glass/epoxy laminate, the same concept can be used to analyze the AE results.
Reviewer 3 Report
The conducted work “Acoustic emission-based analysis of damage mechanisms in filament wound fiber reinforced composite tubes” is good. However, following comments should be addressed to further improve the paper:
A. GENERAL COMMENTS ON PAPER
1. Explicitly mention the novelty and research significance of current work in last paragraph of introduction section with emphasis on scientific soundness. Also, it would be great to add recent relevant literature review from 2023 papers in introduction section as there is only one paper cited from 2023.
2. Avoid paragraph of few (i.e. 2-4) sentences throughout the manuscript.
3. There are too many equations in main text. Only very important equations may be retained in main text, and important equations may be presented in annexure.
4. Texts within figures must be readable.
5. There are too many figures. Only very important figures may be retained in main text body. And important figure may be shown in annexure. Few similar figures can be merged in one figure as a and b. For example, figures 9 and 10 can be shown as a and b. Figures 14 and 15 can also be shown as a and b.
6. Results are very briefly explained/elaborated in a descriptive way, thus results in current form look like a project report. Results should be further elaborated with scientific reasoning.
7. A separate brief section (explaining the relevance of this research for practical implementation) may be added before conclusion section.
8. Conclusions are little long; these should be to the point as obtained from results with scientific soundness. Closing remarks should be added at the end of conclusion section keeping in mind all conclusive bullet points.
9. English Language should be improved throughout the manuscript.
B. SPECIFIC COMMENTS FOR IMPROVING FOCUSSED RESEARCH
1. Figure 3: samples S1 and S2 are showing similar behavior, whereas samples S3 and S4 are showing similar behavior especially between loads 100-150 N and displacement 20-30 mm. Please comment on this.
2. Figure 3: samples P2 and S4 are showing similar behavior, whereas samples P1and P3 are showing similar behavior especially between loads 125-150 N and displacement 15-20 mm. Please comment on this.
1. Avoid paragraph of few (i.e. 2-4) sentences throughout the manuscript, particularly in results section.
Author Response
- GENERAL COMMENTS ON PAPER
- Explicitly mention the novelty and research significance of current work in last paragraph of introduction section with emphasis on scientific soundness. Also, it would be great to add recent relevant literature review from 2023 papers in introduction section as there is only one paper cited from 2023.
The novelty of the work is emphasized in the abstract, page 1, line 11-12, and introduction, page 3, line 123-129.
New references are included.
- Avoid paragraph of few (i.e. 2-4) sentences throughout the manuscript.
It is modified.
- There are too many equations in main text. Only very important equations may be retained in main text, and important equations may be presented in annexure.
The equations are reduced.
- Texts within figures must be readable.
It is modified.
- There are too many figures. Only very important figures may be retained in main text body. And important figure may be shown in annexure. Few similar figures can be merged in one figure as a and b. For example, figures 9 and 10 can be shown as a and b. Figures 14 and 15 can also be shown as a and b.
It is modified.
- Results are very briefly explained/elaborated in a descriptive way, thus results in current form look like a project report. Results should be further elaborated with scientific reasoning.
Some further discussion is included, page 10, lines 320-322; page 11, lines 340-344; page 20, lines 476-478; page 22, lines 492-498.
- A separate brief section (explaining the relevance of this research for practical implementation) may be added before conclusion section.
Some further discussion is included, page 23, lines 554-564.
- Conclusions are little long; these should be to the point as obtained from results with scientific soundness. Closing remarks should be added at the end of conclusion section keeping in mind all conclusive bullet points.
Some sentences were shortened.
- English Language should be improved throughout the manuscript.
It is revised.
- SPECIFIC COMMENTS FOR IMPROVING FOCUSSED RESEARCH
- Figure 3: samples S1 and S2 are showing similar behavior, whereas samples S3 and S4 are showing similar behavior especially between loads 100-150 N and displacement 20-30 mm. Please comment on this.
- Figure 3: samples P2 and P4 are showing similar behavior, whereas samples P1 and P3 are showing similar behavior especially between loads 125-150 N and displacement 15-20 mm. Please comment on this.
These are attributed to defects and issues that occurred during the filament winding manufacturing process and hand-layup process for the reinforcement. It is reflected in the revised version, page 10, lines 320-322 and page 11, lines 340-344.
- Avoid paragraph of few (i.e. 2-4) sentences throughout the manuscript, particularly in results section.
It is modified.
Reviewer 4 Report
I have below comments for this paper:
1-How the authors assumed the interaction force between fibers and matrix?
2-What is the effect of boundary condition on the results?
3-The introduction section can be improved:
Composite Structures 150 (2016) 255-265; Computers and Concrete, An International Journal 21 (2018), 431-440; Aerospace Science and Technology 98 (2020) 105656; European Journal of Mechanics-A/Solids 82 (2020) 104010;
4-Please clearly state the limitations and assumptions of your model
I have below comments for this paper:
1-How the authors assumed the interaction force between fibers and matrix?
2-What is the effect of boundary condition on the results?
3-The introduction section can be improved:
Composite Structures 150 (2016) 255-265; Computers and Concrete, An International Journal 21 (2018), 431-440; Aerospace Science and Technology 98 (2020) 105656; European Journal of Mechanics-A/Solids 82 (2020) 104010;
4-Please clearly state the limitations and assumptions of your model
Author Response
This reviewer's comments are irrelevant and not connected to the content of our paper.
Round 2
Reviewer 2 Report
Some additional comments to the authors' responses are given in red. Please carefully check it.
Yes, other parameters such as energy, duration and count are also considered for clustering, but the best outcome, with the least overlap, was achieved by Frequency and Amplitude distributions. The average frequency used to cluster the data using the SOM algorithm. These are included in the revised version, page 20, lines 476-478.
Standard for a good outcome of clustering analysis is not with the least overlap, but with the best logical understandable compared to the physical evolution of damage. The choose of input feature for clustering analysis could be based on the correlative coefficient. The lower coefficient between two AE features indicates they are independent of each other and therefore could be used as combined features for clustering analysis. Again, only when one outcome is well consistent with the evolution of different damage processes, it can be evaluated to be a good one.
- Better check the time distribution of each cluster with analysis of the deformation and damage evolution during the whole test.
It would have been good to add the suggested information, however, unfortunately, the raw data source we had is no longer accessible due to some problems in the IT system.
The time evolution of different clusters is a simple yet effective way to validate the outcome of the clustering results. But, anyway, it is such a pity with an interpretation like this.
- It is surprising that no microstructure characterization of fracture or microdamages, which were used to explain the AE signals.
The correlation was done considering previous studies in laminated composites where high amplitude and Frequency signals represent fibre fractures while medium values mostly correspond to delamination and interfacial debonding [3, 18-21]. Since those works were related to a similar type of glass/epoxy laminate, the same concept can be used to analyze the AE results.
Although the results are consistent with previous studies. No one figure showing the morphology of microdamage or microfracture is unacceptable.
Author Response
Standard for a good outcome of clustering analysis is not with the least overlap, but with the best logical understandable compared to the physical evolution of damage. The choose of input feature for clustering analysis could be based on the correlative coefficient. The lower coefficient between two AE features indicates they are independent of each other and therefore could be used as combined features for clustering analysis. Again, only when one outcome is well consistent with the evolution of different damage processes, it can be evaluated to be a good one.
We agree with the reviewer. This is indeed considered based on the background knowledge we had about the expected damage evolution and looking at previous works in this area. Several related studies have yielded similar outcomes by employing Principal Component Analysis (PCA) to certain acoustic parameters [references 19, 21, and 23]. These investigations also utilized amplitude and frequency as the fundamental elements for clustering purposes.
Although the results are consistent with previous studies. No one figure showing the morphology of microdamage or microfracture is unacceptable.
We agree with the reviewer. We somehow validated the work with the literature. Unfortunately, the student working on this project is already graduated and is no longer available to carry out the fracture SEM imaging, and employing someone else to do the job is not an option due to its time-consuming nature and lack of resources.
Reviewer 4 Report
Answer to comments is need.
Answer to comments is need.
Author Response
1-How the authors assumed the interaction force between fibers and matrix?
In this study, a mesoscopic modelling approach is used to analyze laminated composites. This method relies on the well-established Classical Lamination Theory (CLT), making it unnecessary to explicitly define the interaction force between the fibres and the matrix.
2-What is the effect of boundary condition on the results?
This is provided in the FEM results section, page 12, lines 381-386, page 13, lines 407-411, and page 16, lines 420-426.
3-The introduction section can be improved:
Composite Structures 150 (2016) 255-265; Computers and Concrete, An International Journal 21 (2018), 431-440; Aerospace Science and Technology 98 (2020) 105656; European Journal of Mechanics-A/Solids 82 (2020) 104010;
The introduction has been revised. Thanks for the suggestions. The introduction section is already large and the suggested papers are quite generic and not focused on the main objective of this work, so respectfully, we decided not to include these references.
4-Please clearly state the limitations and assumptions of your model
Some further discussion is included, page 6, lines 231-233 and page 6, lines 242-244.
Round 3
Reviewer 2 Report
Although the results are consistent with previous studies. No one figure showing the morphology of microdamage or microfracture is unacceptable.
We agree with the reviewer. We somehow validated the work with the literature. Unfortunately, the student working on this project is already graduated and is no longer available to carry out the fracture SEM imaging, and employing someone else to do the job is not an option due to its time-consuming nature and lack of resources.
It takes only 1/2 day to take normal SEM pictures. I don't understand why the author describes it as time-consuming. I just don't want to complain too much. But this is not a scientific discussion based on scientific standards.
The direction is just wrong.
OK, That's all.
Author Response
It takes only 1/2 day to take normal SEM pictures. I don't understand why the author describes it as time-consuming. I just don't want to complain too much. But this is not a scientific discussion based on scientific standards.
The direction is just wrong.
OK, That's all.
It is indeed unfortunate we can not carry the SEMs. It is not just the time that takes to scan, but there is a long queue for the SEM device, and currently, the student is graduated, so it would be a lot of waiting before we can get some SEMs done.
Reviewer 4 Report
Accept
Author Response
The reviewer has accepted the current version.